# Global Potential Geographical Distribution of the Southern Armyworm (*Spodoptera eridania*) under Climate Change

**DOI:** 10.3390/biology12071040

**Published:** 2023-07-23

**Authors:** Yu Zhang, Haoxiang Zhao, Yuhan Qi, Ming Li, Nianwan Yang, Jianyang Guo, Xiaoqing Xian, Wanxue Liu

**Affiliations:** 1State Key Laboratory for Biology of Plant Diseases and Insect Pests, Institute of Plant Protection, Chinese Academy of Agricultural Sciences, Beijing 100193, China; zhangyuu960606@163.com (Y.Z.); hx_zhao@bjfu.edu.cn (H.Z.); qyh_nwnu@163.com (Y.Q.); l1763650355m@163.com (M.L.); yangnianwan@caas.cn (N.Y.); guojianyang@caas.cn (J.G.); 2Institute of Western Agriculture, Chinese Academy of Agricultural Sciences, Changji 831100, China

**Keywords:** southern armyworm, MaxEnt, potential geographical distribution, climate change

## Abstract

**Simple Summary:**

The invasion of *Spodoptera eridania* and the damage it has caused to crop systems in recent years has raised concerns about its potential risks. In this study, we aimed to map the invasion risk of *S. eridania* worldwide based on an optimized MaxEnt model. The results indicated the suitable habitat concentrated in southern North America, South America, western Europe, central Africa, southern Asia, and eastern Oceania. Under climate change, suitable habitats for *S. eridania* will expand and shift to higher latitudes in the future. Countries without *S. eridania* invasion but that are at invasive risk, such as the European Union, Southeast Asian countries, and Australia should take appropriate quarantine measures to prevent the entry of this pest.

**Abstract:**

The southern armyworm (*Spodoptera eridania*), a polyphagous crop pest native to tropical America, has been found in Africa (2016) and India (2019), causing defoliation and damage to the reproductive structures of cassava, soybean, and tomato. The damage caused by this pest to crop systems has raised concerns regarding its potential risks. Therefore, we predicted the potential geographical distribution of *S. eridania* under climate change conditions using 19 bioclimatic variables based on an optimized MaxEnt model. The results showed that annual precipitation (bio12), mean temperature of the warmest quarter (bio10), and precipitation of the driest month (bio14) were important bioclimatic variables influencing the potential distribution. The prediction showed that the suitable habitat area was approximately 3426.43 × 10^4^ km^2^, mainly concentrated in southern North America, South America, western Europe, central Africa, southern Asia, and eastern Oceania. In response to global climate change, suitable habitats for *S. eridania* will expand and shift to higher latitudes in the future, especially under the SSP5-8.5 scenario. Because of the current devastating effects on crop production, countries without *S. eridania* invasion, such as the European Union, Southeast Asian countries, and Australia, need to strengthen phytosanitary measures at border ports to prevent the introduction of this pest.

## 1. Introduction

Invasive crop pests have posed a threat to global agricultural production and food security, as they can reduce the yield and the value of crops and cause economic losses [1]. Recent years, *Spodoptera* species appear to be significant invasive pests globally. This genus consists of more than 30 species, approximately half of which are economically important pests such as *Spodoptera frugiperda* (the fall armyworm), *S. exigua* (the beet armyworm), and *S. exempta* (the African armyworm) [2]. These pests can cause significant economic damage to Gramineae, Brassicaceae, and Fabaceae crops [3,4,5]. Their highly polyphagous nature allows them to adapt rapidly to different agroecosystems [6,7]. *Spodoptera eridania* (Stoll, 1781) (Lepidoptera: Noctuidae) (the southern armyworm, SAW), another significant invasive crop pest of *Spodoptera*, has expanded its range across multiple continents resulting in defoliation and damage to the reproductive structures of plants [8]. It is native to the American tropics; more recently, it has spread to countries outside its native range. In mid-December 2016, SAW larvae were found in a field in south-eastern Nigeria, causing the severe defoliation of cassava [9]. Subsequently, it was found in Benin, Cameroon, and Gabon [9]. Additionally, in September 2019, it was found for the first time in India, with an infection on soybean crop by the larvae [10]. The spread of SAW appears to be largely human-mediated, and it does not engage in long-distance migrations in the Americas [11]. Additionally, like its congeners, the SAW also has a broad host range, with more than 200 natural host plants in 58 families, such as tomato, soybean, cassava, maize, and cotton [11,12]. It remains a destructive pest of tomatoes in Florida [13]. In addition, the SAW can select weeds as a “green bridge” to alleviate the stress caused by temporary food shortages [14].

As a complete metamorphosis insect, the SAW undergoes four stages in its life cycle, egg, larva, pupa, and adult, and is extremely vulnerable to environmental changes. Evidence demonstrates that temperature can directly influence the development rate and survival of each stage [15,16]. The SAW has wide adaptability to temperature; it can complete its development between 15 °C and 32 °C [16], and adults can survive for several days even if the temperature falls below freezing [17]. The number of generations for SAWs is estimated at four annually in Florida with 30 to 40 days for one complete generation [8], and it was predicted that under global climate change, the SAW may reach up to 12 annual generations by 2070 in certain regions [15]. Precipitation also plays an important role in the population development of moths, directly influencing pupa survival and adult emergence [18,19]. SAW larvae pupate in the soil at a depth of 5–10 cm, and the pupa period is about 11 to 13 days, during which soil humidity is required [8]. Furthermore, other environment factors, including extreme events such as rainfalls, are likely to produce outbreaks in unexpected places [20]. Additionally, meta-analysis suggests that climate change can alter the distribution pattern of invasive insects, shifting toward higher elevations and latitudes [21]. In general, the crop pests are expected to benefit from climate change. Considering the highly omnivorous nature and environmental adaptability of the SAW, understanding the potential geographical distribution is of great relevance for its global preventive efforts, especially in the context of global climate change.

Species distribution models (SDMs) are predictive tools that express the effects of environmental change on species distribution [22]. MaxEnt [23] is one of the most popular prediction models and is widely used to predict the potential geographical distribution of invasive pests in risk analysis as it requires only occurrence and environmental data [24,25]. However, the accuracy of the model depends on the quality and quantity of input data; therefore, sample bias from occurrence records may lead to inaccurate models [26,27]. In addition, the complexity of the model plays an important role in inferring habitat quality and the importance of variables in constraining the species’ distribution [28,29]. Therefore, an optimized model with good performance is required to define the extent of invasive alien species under climate change.

In this study, we used an optimized MaxEnt model to map the invasion risk of the SAW worldwide under near-current and future climate scenarios as follows: (1) the significant bioclimatic variables affecting the potential geographical distribution; (2) potential geographical distribution under the near-current climate; (3) the climate change impact on potential geographical distribution; and (4) the spatial variation of potential geographical distribution. This study aims to serve as a valuable reference for preventing SAW invasion worldwide.

## 2. Materials and Methods

### 2.1. Data Access and Processing

Global occurrence data (including coordinates) and distribution region information of the SAW were obtained from the Global Biodiversity Information Facility (GBIF; https://www.gbif.org/, accessed on 23 November 2022), the Center for Agriculture and Bioscience International (CABI; https://www.cabi.org/, accessed on 23 November 2022), Bold Systems (https://www.boldsystems.org/, accessed on 23 November 2022), and the published literature [10,14,30]. After deduplication and correction, 383 occurrence records were considered valid (Figure 1). These occurrence data were assigned to 9 × 9 km climate data grids in ArcGIS 10.2, and only one record was left per grid to minimize sample bias. Finally, 247 occurrence points were retained for model construction and prediction.

Climate data were downloaded from the World Climate Database (https://www.worldclim.org/, accessed on 23 November 2022), including 19 bioclimatic variables (Table 1) with a spatial resolution of 5 min. The near-current conditions were averaged for the years 1970–2000. For the future climate scenarios (2030 scenarios were averages of those from 2021–2040, and 2050 scenarios were averages of those from 2041–2060), we selected two shared socioeconomic pathways, representing the lowest (SSP1-2.6) and highest (SSP5-8.5) scenarios of greenhouse gas emissions, assessed using the global climate model (GCM; BCC-CSM2-MR). To avoid model overfitting and reduce the impact on prediction caused by multicollinearity among the bioclimatic variables, we sampled the environmental data based on 247 occurrence points, and principal component analysis (PCA) (Table 1) and correlation analysis were conducted using IBM SPSS Statistics version 25 to select the variables. When the absolute value of the Pearson correlation coefficient was >0.8, the variable with the highest contribution was retained. Finally, seven bioclimatic variables were selected for model construction (Table 1).

### 2.2. Model Construction and Evaluation

MaxEnt (version 3.4.1) was used to predict the potential geographical distribution of SAW under current and future climate scenarios. The complexity of the MaxEnt model is closely related to that of the regularization multiplier (RM) and feature combination (FC) parameters. The ENMeval package was used to evaluate the complexity of the model by testing the Akaike information criterion correction (AICc) value corrected by the MaxEnt model under different RM and FC parameter combinations [31]. In this study, we built the model with RM values ranging from 0.5 to 6 (in increments of 0.5) and with different FC combinations (H, L, LQ, LQH, and LQHP, where H = hinge, L = linear, Q = quadratic, and P = product). The parameter combination corresponding to the minimum AICc value (AICc = 0) was selected for model prediction [28]. The other model parameters were set as follows: 25% random test percentage, 5000 maximum iterations, 10th percentile training presence threshold rule, 10 replicates, and subsample replicated run type [32].

The model’s performance was evaluated using the area under the curve (AUC) of the receiver operating characteristic (ROC). The value of the AUC ranged from 0 to 1; the higher the value, the better the model performance [33].

### 2.3. Mapping the Potential Geographical Distribution and Spatial Variation

The prediction results from MaxEnt were imported into ArcGIS and converted into raster files. The suitability of the SAW ranged from 0 to 1. According to the maximum test sensitivity plus specificity logistic threshold (0.1254), the reclassify tool of ArcGIS was used to divide the suitability into four fitness classes: “unsuitable”, “low”, “moderate”, and “high”.

The “Calculate Geometry” tool in the attribute table of ArcGIS was used to calculate the area for the suitable habitat in the projected coordinate system and the latitude and longitude of the centroid of the potential geographical distribution in the geographic coordinate system under different climate scenarios.

The analysis of spatial variation in the potential geographic distribution was achieved in ArcGIS and was shown as “unchanged”, “expansion”, and “contraction”.

## 3. Results

### 3.1. Model Performance and Significant Variables

The model optimization results showed that when RM = 2 and FC = LQH, the corresponding AICc value was the lowest (AICc = 0) (Appendix A). Furthermore, the average AUC value of the optimized model was 0.957 with 10 replicates (Appendix A), indicating that our optimized model performed excellently in predicting the potential geographical distribution of the SAW.

Jackknife results showed that the important bioclimatic variables affecting the potential geographical distribution of the SAW were the mean temperature of warmest quarter (bio10), annual precipitation (bio12), and precipitation of driest month (bio14) (Appendix A). The environmental variable with the highest gain was bio12, which appeared to have the most useful information by itself. The response curves of these three important bioclimatic variables shown in Appendix A reflect the dependence of the predicted suitability of the SAW on the selected variable. For the precipitation variable bio12, SAW suitability was positively correlated with bio12 in the range of approximately 200–1342 mm and negatively correlated when it was greater than 1342 mm, with suitability being highest when precipitation reached 1342 mm. For the precipitation variable bio14, SAW suitability was positively correlated with bio14 in the range of approximately 0–80 mm and negatively correlated when it was greater than 80 mm, with suitability being highest when precipitation reached 80 mm. For the temperature variable bio10, SAW suitability was positively correlated with bio10 in the range of approximately 10–27.8 °C and negatively correlated when it was greater than 27.8 °C, with suitability being highest when the temperature reached 27.8 °C.

### 3.2. Potential Geographical Distribution under Near-Current Climate Conditions

The potential geographical distribution of the SAW under the near-current climate is shown in Figure 2. The suitable habitats covered 3426.43 × 10^4^ km^2^ and were mainly located in central and northern South America, central Africa, southern Asia, southeastern North America, eastern Oceania, and western Europe. The corresponding suitable habitat areas on each continent and their proportion of the total risk area were 1213.39 × 10^4^ km^2^ (35.41%), 703.87 × 10^4^ km^2^ (20.54%), 651.43 × 10^4^ km^2^ (19.01%), 456.04 × 10^4^ km^2^ (13.31%), 252.24 × 10^4^ km^2^ (7.36%), and 149.46 × 10^4^ km^2^ (4.36%) (Table 2).

The high suitability habitats for the SAW were mainly located in the south-eastern USA, eastern Mexico, Cuba, and Puerto Rico (North American regions); central Bolivia, southwestern and eastern Brazil, southeastern Paraguay, northeastern Argentina, and northwestern Uruguay (South American regions), eastern Madagascar (Africa), central Vietnam, the Philippines, southern Pakistan (Asia), and southwestern Papua New Guinea (Oceania) (Figure 2).

### 3.3. Climate Change Impact on the Potential Geographical Distribution

The potential geographical distribution of the SAW under future climate scenarios is shown in Figure 3. An overall increasing trend was observed in the area of potential suitable habitats, with the largest area under the SSP5-8.5 scenario in the 2030s (Table 2). The SAW had the largest area of suitable habitats in South America and the smallest area in Europe. Under future climate scenarios, the area of suitable habitats increased in North America, Europe, and Asia, decreased in Africa, and remained largely unchanged in Oceania.

From a suitability perspective, the high-suitability habitat area increased in the 2050s, mainly in China, the USA, and Brazil, and remained largely unchanged in the 2030s. The moderate-suitability habitat area increased under the four conditions, with the overall increase being similar, mainly located in Europe (France), Asia (India), and Oceania (Australia and New Zealand). The low-suitability habitat area was increased under the scenario SSP5-8.5 in the 2030s and SSP1-2.6 in the 2050s, mainly located in South America (Brazil) and Asia (China), and was decreased under SSP1-2.6 in the 2030s and SSP5-8.5 in the 2050s, mainly located in Africa (Chad and South Sudan) (Figure 3 and Appendix A).

### 3.4. Spatial Variation of Potential Geographical Distribution

Analysis of spatial changes in potential geographical distribution allows for a more intuitive representation of the effects of climate change on the potential habitats of species. In this work, the spatial variation in the potential geographical distribution of the SAW under different future climates was expressed as “unchanged”, “expansion”, and “contraction”, compared to that under near-current conditions (Figure 4, Appendix A).

Under future climate conditions, the expansion of suitable habitats was mainly located in North America (the USA, Canada, and Mexico), South America (Brazil and Argentina), Europe (Spain, France, UK, Germany, and Austria), Africa (Madagascar, Ethiopia, the Democratic Republic of the Congo, and South Africa), Asia (China, Myanmar, Republic of Korea, Japan, Indonesia, and Thailand), and Oceania (Australia and Papua New Guinea). The greatest expansion of suitable habitat occurred in the 2050s under the SSP5-8.5 scenario, and the expansion area was 464.40 × 10^4^ km^2^.

The contraction area indicates that under future climatic conditions, this habitat will no longer be suitable for the SAW, and the habitat will be lost. For the SAW, the largest area of habitat loss (397.94 × 10^4^ km^2^) occurred under the SSP5-8.5 climate scenario in the 2050s, mainly in northern South America, central Africa, and Asia (India and Thailand).

In summary, under the SSP5-8.5 scenario in the 2050s, the spatial variation in potential geographical distribution for the SAW was the greatest, with the “expansion”, “contraction”, and “unchanged” areas being 464.40 × 10^4^ km^2^, 397.94 × 10^4^ km^2^, and 3028.49 × 10^4^ km^2^, respectively.

The centroid of the potential geographical distribution was analyzed to assess the migration of this species’ suitable habitats in latitudinal and longitudinal coordinates under the influence of climate change. In this study, the potential geographical distribution centroids of the SAW in each continent under different climate scenarios were calculated (Appendix A) and are shown in Figure 5. In North America and Europe, the centroids of suitable habitats shifted northward and were located in the USA and France, respectively. In Asia, the centroid of suitable habitat was located in Thailand (under near-current and SSP5-8.5 in the 2030s) and shifted to Laos under other scenarios; this was an overall shift northward. In South America, Africa, and Oceania, the centroid of suitable habitats was located in Brazil, the Congo, and Australia, respectively, and shifted southward.

In summary, compared with that in the near-current climate, the centroid of suitable habitats under future climate scenarios for the SAW shifted to higher latitudes, indicating that the potential geographical distribution of the SAW would occupy a wider latitudinal range under the impact of climate change.

## 4. Discussion

The SAW threatens the agroecosystem due to its polyphagous nature and environmental adaptability. In this study, we explored the impact of changes in temperature and precipitation on the potential geographical distribution of this invasive species using the MaxEnt model.

### 4.1. Model Prediction

In recent years, the CLIMEX and MaxEnt models have been widely used to explore the relationship between distribution patterns and climate [34]. Both models have been used previously to predict the potential distribution of the SAW [35,36]. The CLIMEX results showed that the SAW had a wide suitable habitat area in Africa, while Europe was basically not suitable [35], which is different from the results of our model. The suitable habitat area in North America was smaller than that in our results, especially in southeastern USA. Compared with the results from Tepa-Yotto’s prediction [36], there was a difference in the suitable habitats on each continent, especially in South America, Southeast Asia, and Europe. The area of suitable habitats was smaller than the predicted result of our model. The difference between the prediction results in these three models may be due to many reasons. Firstly, we considered the occurrence data reported in India, while the previous studies did not, and a larger number of occurrence points were collected in our model. Secondly, the model’s parameter settings and the processing of the resulting data (e.g., criteria for delineating suitable habitats) directly impact the presentation of prediction results. Although there were differences between the previous studies and our study, all prediction results indicated that suitable habitats for the SAW would expand under future climate conditions. In addition, we explored the spatial variation of suitable habitats, which expressed a more intuitive representation of the effects of climate change on the potential geographical distribution of the SAW and provided a novel reference for the prevention and management of the SAW worldwide, especially for the countries that have not yet been invaded.

### 4.2. Influence of Bioclimatic Variables on Potential Geographical Distribution

Temperature plays an important role in the development of insects and can directly influence their life cycle, especially extreme temperatures, and determine their potential geographical range [18,20]. Our findings indicated that the mean temperature of the warmest quarter (bio10) was an important variable affecting the potential geographical distribution of the SAW; the suitable mean temperature of the warmest quarter for the SAW was approximately 25.3–28.5 °C, and when the temperature was below 10 °C or above 33 °C, the suitability to the SAW tended towards zero. This finding was supported by previous studies on the effect of temperature on SAW development [37]. The intolerance of eggs and larvae to extremely high temperatures [16,37] may restrict its establishment or lead to low suitability in tropical regions, such as western and northern Africa and India. Global warming and wide-range adaptation to temperature would drive the SAW towards higher latitudes.

Precipitation can, directly and indirectly, influence the spread and population establishment of invasive crop pests [19,20]. For the SAW, precipitation can directly influence pupal survival and adult emergence. In additional, precipitation can indirectly affect the distribution of the SAW by influencing the distribution of host plants. In this study, we found that the annual precipitation (bio12) and the precipitation of the driest month (bio14) were the key variables affecting the potential distribution of SAWs. When the annual precipitation was less than 500 mm or the precipitation of driest month was approximately 0 mm, the suitability to the SAW tended towards zero, indicating that the SAW was intolerant to drought and its distribution was restricted to arid regions. Further studies on the influences of soil type and rainfall on SAW survival are required for effective management and control.

### 4.3. Changes in Potential Geographical Distribution

The results of this study suggest that SAWs have a wide range of suitable habitats in areas outside their origin. Regarding the impact of climate change on potential suitable habitats, SAWs had the largest suitable habitat under SSP5-8.5 in the 2030s, suggesting that this environmental condition is more suitable for the spread and development of this species’ population than other climatic conditions are. In future climate conditions, the SAW-suitable habitats will expand towards high latitudes; however, certain habitats in Africa will be lost, which is consistent with previous studies [36]. This may be due to the limitations of precipitation and extreme heat, as discussed above and in the previous study [35].

Global climate change, particularly global warming, has facilitated the spread and establishment of many invasive species at higher latitudes [21]. After analyzing the centroid of the potential suitable habitats for the SAW, we observed that it shifted towards higher latitudes, consistent with the findings of previous studies [21]. In addition, we observed the greatest variation in the potential suitable habitats of the SAW under SSP5-8.5 in the 2050s, mainly in terms of the largest area of expansion and contraction. Combined with the fact that the suitable habitat spanned a wider latitude, and that the area of the suitable habitat did not change much, we conclude that the fragmentation of suitable habitats for the SAW increased under future climate scenarios, which could directly affect the spread and development of SAW populations [38,39].

In summary, climate change can alter the potential distribution of SAW. Global warming and adaptation to a wide range of temperatures allow the SAW to occur in regions outside its native range and move to higher latitudes, whereas the requirements of precipitation and intolerance to extreme heat cause a loss of its habitat in future conditions. 

### 4.4. Spread and Prevention Efforts

Invasive crop pests can spread through natural and human means. In particular, human-mediated dispersals, such as international trade and travel, allow species to overcome long distances and geographical barriers to be introduced into non-infested areas globally; globalization is likely to exacerbate this problem [40]. A previous study has mentioned that the SAW does not engage in long-distance migrations in the Americas, and human-assisted dispersal is the likely cause of its spread to Africa [11]. Therefore, the countries with suitable conditions for the SAW that have not yet been invaded, especially China and the EU, which are major importers of tomato, soybean, and cassava (host plants of SAWs) [41,42,43], should pay special attention to the global trade of host vectors, which can transport larvae, eggs, and pupae. The EU banned the import of soil from countries where the SAW occurs to prevent the entry of pupae [11] and listed this species as an A1 Quarantine pest in 2019 [44]. In China, there are appropriate environmental conditions for its survival and reproduction [45], and hosts are widely cultivated, providing sufficient food, especially in southern China, where it is more suitable for the SAW under climate change. Suitable habitats for SAWs in China are expected to expand northward. Once established, it will pose a threat to agriculture in China. Therefore, countries at invasive risk should take appropriate quarantine measures and conduct the targeted testing of imported plants and their products to prevent the introduction of this pest. Establishing more effective inspection systems and identification technologies at international borders is essential for identifying this pest and disrupting its introduction routes. In addition, as there is often a time lag between initial introduction and observation (such as in India and Africa, where the population was not noticed until a large outbreak occurred), a complete surveillance system is required to detect this pest early, and the highly polyphagous nature of the SAW means that the management of this pest should focus on wide crop systems.

## 5. Conclusions

We used the global occurrence records of the SAW and an optimized MaxEnt model to predict its potential geographical distribution under climate change. Our study concluded that the SAW has a widely suitable habitat outside its native range under the near-current climate, and it is expected to become wider in the future, although its distribution is restricted. Annual precipitation is the most important bioclimatic variable affecting the potential distribution of the SAW. The spatial distribution of the potentially suitable habitats will span a wider latitudinal range under climate change in the 2030s and 2050s. In summary, the SAW poses a threat to agricultural development in native and invasive regions, and environmental changes exacerbate this situation. Invasion hotspots, such as Australia, southern China, the EU, and other countries that are not yet invaded but are suitable for the SAW should adopt scientific quarantine measures to prevent the entry of this pest.

## Figures and Tables

**Figure 1 biology-12-01040-f001:**
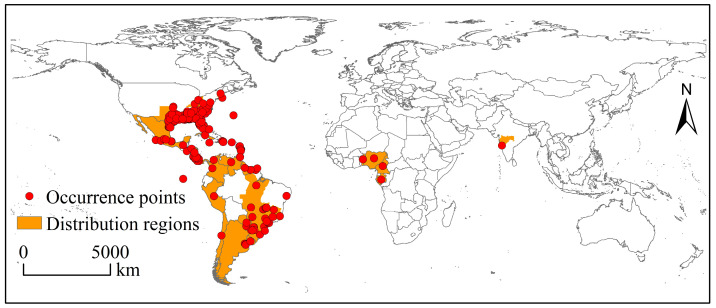
The occurrence data and distribution region of *Spodoptera eridania* around the world.

**Figure 2 biology-12-01040-f002:**
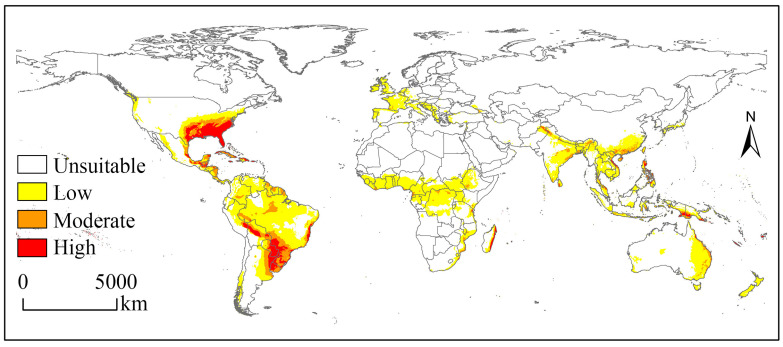
Potential geographical distribution of *Spodoptera eridania* under near-current climate.

**Figure 3 biology-12-01040-f003:**
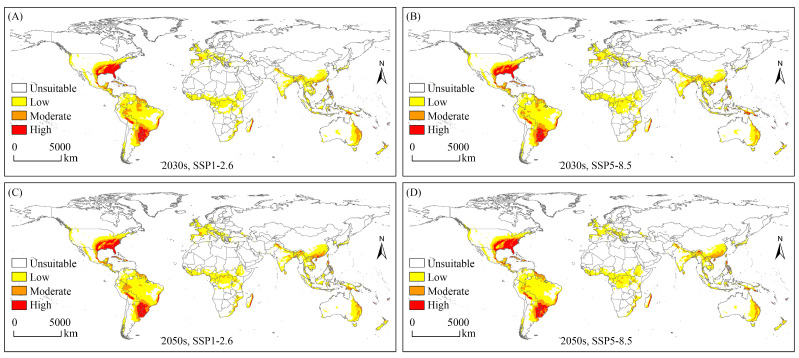
Potential geographical distribution of *Spodoptera eridania* under future climate scenarios. (**A**) Potential geographical distribution of *Spodoptera eridania* under the SSP1-2.6 scenario in the 2030s; (**B**) Potential geographical distribution of *Spodoptera eridania* under the SSP5-8.5 scenario in the 2030s; (**C**) Potential geographical distribution of *Spodoptera eridania* under the SSP1-2.6 scenario in the 2050s; (**D**) Potential geographical distribution of *Spodoptera eridania* under the SSP5-8.5 scenario in the 2050s.

**Figure 4 biology-12-01040-f004:**
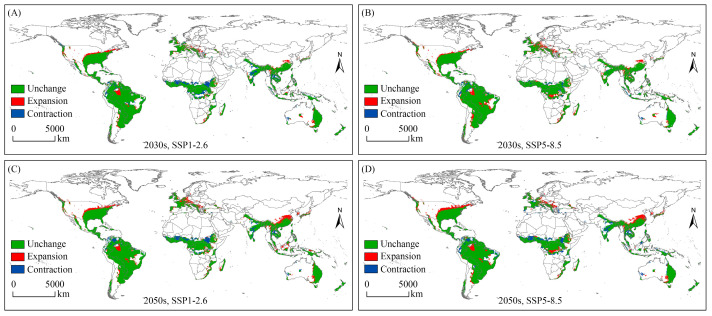
Spatial variation of potential geographical distribution for *Spodoptera eridania* under future climate scenarios. (**A**) Spatial variation of potential geographical distribution of *Spodoptera eridania* under the SSP1-2.6 scenario in the 2030s; (**B**) Spatial variation of potential geographical distribution of *Spodoptera eridania* under the SSP5-8.5 scenario in the 2030s; (**C**) Spatial variation of potential geographical distribution of *Spodoptera eridania* under the SSP1-2.6 scenario in the 2050s; (**D**) Spatial variation of potential geographical distribution of *Spodoptera eridania* under the SSP5-8.5 scenario in the 2050s.

**Figure 5 biology-12-01040-f005:**
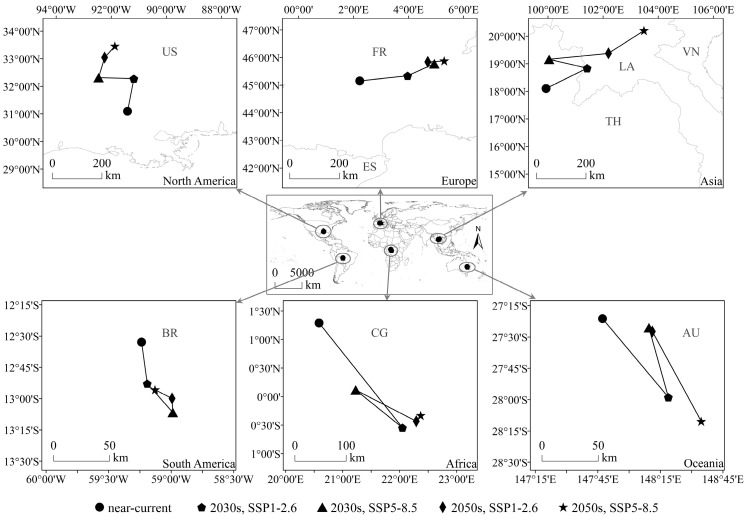
Potential geographical distribution centroid of *Spodoptera eridania* under different climate scenarios. AU: Australia; BR: Brazil; CG: Congo; ES: Spain; FR: France; LA: Laos; TH: Thailand; US: USA; VN: Vietnam.

**Table 1 biology-12-01040-t001:** Principal component analysis (PCA) performed on 19 bioclimatic variables.

Bioclimatic Variables	Principal Components			
	1	2	3	4
Annual mean temperature (bio1)	0.598	−0.319	0.586	0.426
Mean diurnal range (bio2)	−0.428	−0.071	−0.658	0.069
**Isothermality (bio3)**	0.774	−0.344	0.327	−0.270
Temperature seasonality (bio4)	−0.716	0.405	−0.477	0.248
Max temperature of warmest month (bio5)	−0.208	0.020	−0.177	0.935
Min temperature of coldest month (bio6)	0.720	−0.301	0.597	0.074
Temperature annual range (bio7)	−0.705	0.275	−0.586	0.214
**Mean temperature of wettest quarter (bio8)**	0.031	−0.213	0.749	0.376
**Mean temperature of driest quarter (bio9)**	0.775	−0.090	0.110	0.232
**Mean temperature of warmest quarter (bio10)**	−0.066	0.100	0.184	0.963
Mean temperature of coldest quarter (bio11)	0.709	−0.380	0.565	0.099
**Annual precipitation (bio12)**	0.831	0.401	0.168	−0.231
Precipitation of Wettest Month (bio13)	0.883	−0.040	0.194	−0.201
**Precipitation of driest month (bio14)**	−0.042	0.962	0.004	−0.012
Precipitation seasonality (bio15)	0.357	−0.867	0.134	−0.109
Precipitation of wettest quarter (bio16)	0.877	−0.029	0.210	−0.256
Precipitation of driest quarter (bio17)	0.056	0.977	0.004	0.004
Precipitation of warmest quarter (bio18)	0.145	0.302	0.661	−0.273
**Precipitation of coldest quarter (bio19)**	0.725	0.520	−0.218	0.046

Note: Bioclimatic variables in bold were selected for model construction.

**Table 2 biology-12-01040-t002:** The potential geographical distribution area (× 10^4^ km^2^) and percentage (%) of total risk area for *Spodoptera eridania* under near-current and future climate scenarios.

Continents	Near-Current		2030s, SSP1-2.6		2030s, SSP5-8.5		2050s, SSP1-2.6		2050s, SSP5-8.5	
	Area	%	Area	%	Area	%	Area	%	Area	%
Africa	703.87	20.54	581.69	17.25	727.04	19.96	636.83	17.69	645.23	18.47
Asia	651.43	19.01	617.84	18.32	682.25	18.73	729.67	20.27	652.91	18.69
Europe	149.46	4.36	174.82	5.18	197.55	5.42	198.41	5.51	176.72	5.06
North America	456.04	13.31	510.05	15.12	515.25	14.14	535.46	14.88	553.25	15.84
Oceania	252.24	7.36	256.32	7.60	257.31	7.06	265.25	7.37	251.77	7.21
South America	1213.39	35.41	1232.37	36.54	1263.81	34.69	1233.67	34.28	1213.01	34.73
World	3426.43		3373.10		3643.21		3599.29		3492.89	

## Data Availability

The data presented in this study are available in this article.

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
