# Peer review of "Global Potential Geographical Distribution of the Southern Armyworm (Spodoptera eridania) under Climate Change"

_biology, 2023, doi:10.3390/biology12071040_

Round 1
Reviewer 1 Report
- Line 58-61 : How this species spread to Africa and India ?
- Line 194 : Is this result show in table 2 ?
- Line 193-195 : It's unclear. From table 2, it seem like Europe have unchanged.
- Line 215-216 : Line 195 mentioned that the area of suitable habitat remain unchanged.
- Line 222 : Thailand will be no longer suitable or expansion area ? It was mention to be expansion area (Line 216) and contraction area (Line 222)
- Line 256-258 : Can the authors add the reference for this sentence ?
Reviewer 2 Report
In this study, the authors predicted the potential geographical distribution of southern armyworm (Spodoptera eridania), under climate change conditions using 19 bioclimatic variables based on an optimized MaxEnt model. Overall, the topic is interesting and the results support the main conclusion. However, the manuscript needs careful proofreading and revision. Grammar mistakes are undermining the significance of this study. Therefore, I think it cannot be accepted in its current form in insects. I recommend a major revision, in which the following key points should be addressed.
- The introduction section should be revised completely. It should be coherent. Remove the unnecessary and general statements. The authors should focus on target insect, or at least focus on same family or order.
- It would be very interesting, if authors combine the CLIMAX model to MaxEnt, and construct the bivarieable figures. Because just like Maxent, the use of CLIMEX model to predict species distribution patterns has become very common across the academic field.
- The AUC is a classical approach, but it is not without criticism. There are several other metrics that can be used for the model performance evaluation of the statistics-based SDMs, such as TSS, OR, and AIC. I strongly suggest authors to use multiple measures.
- I strongly suggest authors to use PCA to statistically select model variables to prevent multicollinearity among them. Authors should add PCA table in main text file.
- The number of references are 58 which is more than enough for this work. I suggest to reduce it as much as possible, 35-40 will be okay.
- Again, English is the major issue in this manuscript. Please revise the document by a native speaker or professional company. Check whole document for typo mistakes.
English is the major issue in this manuscript. Please revise the document by a native speaker or professional company. Check whole document for typo mistakes.
Reviewer 3 Report
The research has an important value and in line with the new trend.
That study, aimed to map the invasion risk of Southern Armyworm worldwide based on an optimized MaxEnt model. Climate change can alter the potential distribution of SAW, so countries without SAW invasion but at invasive risk, should take appropriate quarantine measures to prevent the entry of this pest.
The abstract is including the whole necessary information
Introduction is well-structured, informative and makes the relational for carrying out the study clear.
Methods are straightforward and easy to follow.
Results are well presented And discussed clearly and concisely.
Figures and maps are well provided
There are no missing references.
Author Response
Dear reviewer: Thank you very much for reviewing this manuscript. Best wishes to you!
Reviewer 4 Report
The research topic of the authors is relevant. The scope of work is large. However, there are some comments on the presented material.
Title. Global Potential Geographical Distribution of the Southern Armyworm (Spodoptera eridania) Under Climate Change
Keywords. Keywords are necessary for search systems, therefore they must not be doubled in the Title.
In the Introduction, it is necessary to add a few phrases about the life cycle of the SAW. Wintering stage. Which stages are vulnerable to adverse weather conditions in the soil, on the food plant, during migration, in which months and/or decades?
line 47: “The genus Spodoptera consists of 31 species” - I suppose it is better to say “more than 30 species”
line 69-70: “Previous predictions indicated that, under global climate change, the SAW may reach up to 12.1 annual generations by 2070 in certain regions [24]” I suppose, you can write «...at least 12 generations».
lines 353-354 “The spatial distribution of the potential suitable habitats” – may be “...potentially suitable habitats”
minor editing
Round 2
Reviewer 2 Report
Thanks to authors for addressing previous comments. After revisions, I think the manuscript can be accepted in this journal. No more comments from my side.